# Alzheimer’s Disease, Brain Injury, and C.N.S. Nanotherapy in Humans: Sonoporation Augmenting Drug Targeting

**DOI:** 10.3390/medsci5040029

**Published:** 2017-11-24

**Authors:** Joseph S. D’Arrigo

**Affiliations:** Cavitation-Control Technology Inc., Farmington, CT 06032, USA; cavcon@ntplx.net

**Keywords:** Alzheimer’s disease, drug targeting, nanoemulsion, neuroinflammation, neurotrauma, oxidative stress, scavenger receptors, SR-BI, transcranial sonoporation

## Abstract

Owing to the complexity of neurodegenerative diseases, multiple cellular types need to be targeted simultaneously in order for a given therapy to demonstrate any major effectiveness. Ultrasound-sensitive coated microbubbles (in a targeted nanoemulsion) are available. Versatile small-molecule drug(s) targeting multiple pathways of Alzheimer’s disease pathogenesis are known. By incorporating such drug(s) into the targeted lipid-coated microbubble/nanoparticle-derived (LCM/ND) lipid nanoemulsion type, one obtains a multitasking combination therapeutic for translational medicine. This multitasking therapeutic targets cell-surface scavenger receptors (mainly scavenger receptor class B type I (SR-BI)), making it possible for various Alzheimer’s-related cell types to be simultaneously sought for localized drug treatment in vivo. Besides targeting cell-surface SR-BI, the proposed LCM/ND-nanoemulsion combination therapeutic(s) include a characteristic lipid-coated microbubble (LCM) subpopulation (i.e., a stable LCM suspension); such LCM substantially reduce the acoustic power levels needed for accomplishing temporary noninvasive (transcranial) ultrasound treatment, or sonoporation, if additionally desired for the Alzheimer’s patient.

## 1. Background—Transcranial Ultrasound

The recent preclinical report of using transcranial ultrasound to clear out amyloid-β plaques [1] in mouse brain is quite intriguing, but a related technical news report [2] questions whether this method can work in people without causing damage. Alzheimer’s patients already have disrupted blood–brain barriers, so that any interaction of microbubbles (acoustically activated by ultrasound) with the blood–brain barrier (BBB) needs to be done very carefully so as not to make matters worse for the Alzheimer’s patient [2]. This expressed caution also has relevance to a recent review concerning therapies for Alzheimer’s disease [3]. The authors summarize the field by emphasizing that many of the therapeutic strategies tested have been successful in animal models, but none have been successful in humans. There is a striking deficit in translational research, i.e., taking a successful treatment in mice and translating it to the Alzheimer’s patient. The authors assert that either the rodent models are not good, or we should extract only the most useful information from those animal models [3]. In view of all the foregoing arguments, it appears likely that intravenous injection of film-stabilized microbubbles is quite useful since such preformed microbubbles are well known to substantially reduce the acoustic power levels needed for temporary noninvasive (transcranial) ultrasound opening of the BBB [4,5,6], that is, for accomplishing “sonoporation”. 

## 2. Sonoporation

The structural mechanism for sonoporation by microbubbles/nanobubbles has very recently been studied [7] in more detail by performing molecular dynamics computer simulations on systems that contain a model of the tight junctions from the BBB. When no bubble is present in the system, no damage to the model tight junction is observed when the traveling shock (or sonic) wave propagates across it. However, in the presence of a nanobubble, even when the impulse of the shock wave is relatively low, the implosion of the nanobubble causes significant structural change to the model tight junction [7]. These investigators further explain the structural mechanism of (lipid-bilayer) membrane poration, from shock wave (or sonic wave) induced nanobubble collapse, through the use of (coarse-grain) molecular dynamics simulations. Specifically, in the absence of a nanobubble, shock pressure is evenly distributed along the lateral area of the (modeled lipid-bilayer) membrane; in the presence of a nanobubble, an unequal distribution of pressure on the membrane is created, leading to the membrane poration [8]. 

## 3. Receptor-Mediated Drug Delivery for Alzheimer’s Disease

Moreover, by appropriate choice of film-stabilized microbubbles that can also carry a suitable drug across the BBB for localized delivery, it may be possible for the ultrasound intensity (acoustic power level) to be lowered even further—resulting in even smaller chances of doing any harm to brain tissue in the patient. In actuality, various types of film-stabilized microbubble agent exist which can function as drug carriers. However, many of these preformed microbubble agents are incapable, after intravenous injection, of targeting any localized tissue sites or specific lesions. While some of the remaining film-stabilized microbubble agents are capable of targeting, very few appear capable of searching out the appropriate (cell-surface) receptors lining the vasculature of the human brain or within Alzheimer’s disease sites in actual patients. Those Alzheimer’s-disease-related human receptors involve certain “lipoprotein receptors”, including, notably, the (class B) scavenger receptor referred to as SR-BI (for a review, see Section 24.2 in ref. [9]) which has been found to display significantly impaired function in Alzheimer’s patients [10]. In this study of humans—where, as in mice [11], the SR-BI is well established as a major high-density lipoprotein (HDL) receptor—HDLs were specifically isolated from 20 healthy subjects and from 39 Alzheimer’s patients. The anti-inflammatory activity of HDL was found to be significantly lower in Alzheimer’s patients, which paralleled additional results revealing that Alzheimer’s disease had impaired the interaction of HDL with SR-BI receptors obtained from these patients. The authors conclude that their study, using humans, provides evidence for the first time that the functionality of HDL is impaired in Alzheimer’s disease, and that this alteration might be caused by Alzheimer’s-disease-associated oxidative stress and inflammation [10]. More recently, Song et al. [12] have similarly showed that the anti-inflammatory effects of HDL are dependent on SR-BI expression on macrophages (a type of immune cell). These investigators point out that, besides HDL’s role in regulating cholesterol metabolism, HDL has been shown to exhibit antioxidant and anti-inflammatory effects in the vasculature [12]. To summarize the various cell types which all display cell-surface SR-BI and are potentially implicated in Alzheimer’s disease, the report by Thanopoulou et al. [11] should next be considered. These authors point out that SR-BI has been identified on astrocytes and vascular smooth muscle cells in Alzheimer’s disease brain, and has been demonstrated to mediate adhesion of microglia (another type of immune cell) to fibrillar amyloid-β. As concerns their own experiments, Thanopoulou et al. report that SR-BI mediates perivascular macrophage response, and regulates amyloid-β pathology and cerebral angiopathy in an Alzheimer’s mouse model (i.e., human-amyloid precursor protein transgenic mouse). The authors remark that these findings designate SR-BI as a therapeutic target for the treatment of Alzheimer’s disease and cerebral amyloid angiopathy [11]. 

From all the foregoing findings in the preceding paragraph, it is evident that choosing an intravenous film-stabilized microbubble agent which targets cell-surface SR-BI could allow various above-described cell types, all potentially implicated in Alzheimer’s disease, to be simultaneously searched out and likely reached for localized treatment (e.g., drug delivery). Due to the complexity of Alzheimer’s disease, it is likely that therapeutics which target multiple cellular sites will result in a more efficient management of this disease, and might also be effective in various forms of Alzheimer’s disease with different underlying pathophysiological mechanisms [13]. As recently pointed out by Bredesen [14], there is not any single drug currently available for Alzheimer’s disease that exerts anything beyond a marginal, unsustained symptomatic effect, with little or no effect on disease progression. Bredesen further states that, in the past decade alone, hundreds of clinical trials have been conducted for treating Alzheimer’s disease, at an aggregate cost of literally billions of dollars, without success. However, for both Alzheimer’s disease as well as its predecessors—mild cognitive impairment and subjective cognitive impairment—comprehensive combination therapies (targeting multiple cellular sites) have not been explored. It is also possible that targeting multiple cellular sites within the multiple-cell-type network underlying Alzheimer’s disease pathophysiology may be successful even when each (SR-BI bearing) cell type targeted is affected in a relatively modest way; that is to say, the effects on the various cell types targeted may be additive, multiplicative, or otherwise synergistic [14].

## 4. Past Targeted Nanotherapy Using Lipid Nanoemulsions

The above-stated desire for a multitasking combination therapeutic capable of targeting (via SR-BI) the multiple-cell-type network underlying Alzheimer’s disease pathophysiology would be further fulfilled if the chosen intravenous microbubble agent could readily and demonstrably carry (one or more) useful small molecular drugs(s). There is one multitasking therapeutic candidate, existing in the form of an intravenous film-stabilized microbubble agent which targets cell-surface SR-BI, that is documented to be a successful carrier of selected small molecular compound(s). Specifically, “lipid-coated microbubble (LCM)/nanoparticle-derived” lipid nanoemulsion, also known as the LCM/ND lipid nanoemulsion type, is well documented [9] as useful for the highly selective delivery of (easily incorporated) lipophilic dyes, labels, or low-molecular-weight drugs to various types of solid tumors and certain other (noncancerous) hyperproliferative-disease lesions/sites. All these lesions consistently display an increased (cell-surface) expression and/or activity of lipoprotein receptors, including, notably, the (class B) scavenger receptor known as SR-BI (or sometimes as CLA-1, the human SR-BI ortholog). Such data on SR-BI expression and function are noteworthy; namely, SR-BI has emerged as the lipoprotein receptor primarily involved in the enhanced endocytosis (i.e., enhanced intracellular uptake) of LCM/ND lipid nanoemulsions into hyperproliferative-disease sites [9]. First, as concerns tumors, an independent evaluation of this type of lipid nanoemulsion has appeared in a review article by Constantinides et al. [15]. At the same time, this particular study provides certain relevant data that are useful as a test of the expectation that a significantly enhanced endocytosis of LCM/ND lipid nanoemulsion (likely mediated by SR-BI) ought to be readily detectable in Hep3B human hepatoma cells. This expectation arises from the fact that SR-BI expression, which is well described for HepG2 cells, has also been documented in Hep3B cells. Furthermore, when studying the effect of chemical agents causing decreased SR-BI levels in Hep3B hepatoma cells, the same chemical agents were observed to cause decreased uptake of HDL lipids into Hep3B cells (for a review see ref. [9]). In actuality, a noticeably enhanced uptake of this (dye-carrying) LCM/ND lipid nanoemulsion type into varied tumor cells is reported by Constantinides et al. [15] and, as expected, the observed enhanced uptake is particularly marked in Hep3B hepatoma cells (see Table 24.1 in ref. [9]; cf. “parent Table” (p. 763) in Adv. Drug Deliv. Rev. (2008) 60:757–767). The LCM/ND lipid nanoemulsion version employed by these authors is called Emulsiphan. Most solid tumors displayed enhanced uptake of this Emulsiphan version of (dye-labeled) LCM/ND lipid nanoemulsion; however, these tumors did not do so to the same degree. Nonetheless, it is noteworthy that all of the varied tumor cells listed in Table 24.1 (of [9]; cf. above) display a significantly increased uptake of this LCM/ND lipid nanoemulsion version (as compared with the undetectable level of Emulsiphan nanoemulsion uptake in parenteral 3T3-L1 cells, which are noncancerous cells) (for added discussion, see Section 24.3 in ref. [9]). Besides the above dye-labeling experiments, both Constantinides et al. [15] and Ho et al. [16] have formulated LCM/ND lipid nanoemulsions with the anticancer drug, paclitaxel, and documented the successful delivery (intracellularly) of the carried drug to tumor cells of various types [9]. 

Of the above-mentioned “certain other (noncancerous) hyperproliferative-disease lesions/sites”, which overexpress scavenger receptors, one example is central nervous system (CNS) injury—that is, brain injury and/or spinal cord injury. Various published studies indicate increased scavenger receptor expression on “proliferating macrophages” and “activated astrocytes” arising after CNS injury. At the same time, this increased scavenger receptor expression, which probably mainly involves SR-BI (see Section 25.1.1 in ref. [9]), provides a plausible avenue for targeted drug-delivery treatment of CNS-injury sites. Accordingly, Wakefield et al. [17] examined the use of LCM/ND lipid nanoemulsion to deliver 7β-hydroxycholesterol (7β-OHC) to a radiofrequency (thermal) lesion in the rat brain. (7β-OHC and other oxysterols have been reported, by other investigators, to inhibit astrogliosis both in vitro and in vivo (cf. [9]). Wakefield et al. [17] observed that the number of activated astrocytes were reduced when treated with 7β-OHC delivered by the LCM/ND lipid nanoemulsion, while not affected by the same dose of intravenously injected 7β-OHC in saline. It appears that the mechanism of this enhanced delivery of 7β-OHC to the brain-injury site, by a LCM/ND lipid nanoemulsion type, shares common features with the above tumor work (for added discussion, see Chapter 13 and Section 24.3 in ref. [9]). The above interpretation of the data receives additional indirect support from published findings, of other investigators, which document the expression of SR-BI on astrocytes and vascular smooth muscle cells in adult mouse and human brains—as well as in Alzheimer’s disease brain [9]. Lastly, this documented ability of LCM/ND lipid nanoemulsion to function as a carrier of selected small molecular compounds would, of course, be potentially applicable to certain drug molecules already being used in research for treating Alzheimer’s disease (and brain injury). Several low-molecular-weight (and sufficiently lipophilic) candidates for incorporation into the LCM/ND lipid nanoemulsion are Edaravone [18,19], caffeine [20,21,22,23], resveratrol [24,25], and docosahexaenoic acid (DHA) [26,27,28,29,30,31,32,33,34].

## 5. Serum Amyloid A, SR-BI, and Alzheimer’s Disease

The immune response after brain injury, and during neurodegenerative disorders, is highly complex—involving both local and systemic events at the cellular and molecular level [35]. More specifically, inflammation of brain tissue in the absence of infection (sterile inflammation) contributes to acute brain injury and chronic disease. Accordingly, Savage et al. have studied the inflammatory responses of glial cells in the presence of a relevant endogenous priming stimulus; these authors report the acute-phase-protein serum amyloid A (SAA) (see below) acted as a sterile, endogenous, priming stimulus on glial cells [36]. Note that serum amyloid A (SAA) is a liver-derived “high-density lipoprotein (HDL)”-associated apolipoprotein, whose level in the blood increases up to 1000-fold in response to various injuries including trauma (e.g., CNS injury), inflammation (e.g., human vascular plaques and Alzheimer’s lesions), etc. Like other acute-phase reactants, the liver is the major site of SAA expression; however, SAA is also expressed in cells at inflammation sites, e.g., macrophage cell lines and within human atherosclerotic lesions (e.g., [9]). Baranova et al. point out [37] that the importance of SAA in various physiological and pathological processes has generated considerable interest in the identity of the cell-surface receptor(s) that bind, internalize, and mediate SAA-induced proinflammatory effects. Furthermore, these authors assert that the results of their study demonstrate that CLA-1 (the human SR-BI ortholog [38]) functions as an endocytic SAA receptor, and is involved in SAA-mediated cell signaling events associated with the immune-related and inflammatory effects of SAA [37]. In addition, CLA-1 and SR-BI are highly expressed on monocytes/macrophages, cells known to be the primary sites of SAA uptake [37,39]. 

It is also worth noting that such blood-borne human monocytes (with their high expression of CLA-1/SR-BI and ability to differentiate into macrophages to elicit an immune response locally) have recently been reported [40] to reduce Alzheimer’s-like pathology and associated cognitive impairments in transgenic mice having Alzheimer’s-like symptoms. Specifically, monocytes (derived from human umbilical cord blood cells) were found to play a central role in ameliorating cognitive deficits and reducing amyloid-β neuropathology in an Alzheimer’s mouse model [40]. This finding is consistent with an earlier study, by different investigators [41], which reported that very old SR-BI knockout mice show deficient synaptic plasticity (long-term potentiation) in the hippocampus. Also, very old SR-BI knockout mice were found to display impairments in recognition memory and spatial memory [41]. 

Returning to the above observations regarding SAA and inflammation, they are of added interest because inflammation is a known risk factor for Alzheimer’s disease and the SAA concentration is much higher, in cerebrospinal fluid (CSF), in subjects with Alzheimer’s disease than in controls [42]. It was further found that SAA dissociated apolipoprotein E (apoE) from HDL, in the CSF, in a dose-dependent manner. Importantly, amyloid-β fragments (i.e., 1-42) were bound to large CSF-HDL, but not to apoE dissociated by SAA. Miida et al. [42] therefore postulate that inflammation in the CNS may impair amyloid-β clearance due to the loss of apoE from CSF-HDL. Moreover, it has recently been independently reported that SAA itself can misfold and potentially lead to systemic amyloidoses [43]. 

## 6. Treating Brain Injury, Neuroinflammation, and Alzheimer’s Disease via LCM/ND Nanoemulsions 

The brief histological description of brain-injury sites, in the preceding four paragraphs, points to a larger pathophysiological overlap which exists between brain injury and Alzheimer’s disease brain. First, as concerns brain injury, Wang et al. [44] have pointed out that non-neuronal brain cells, especially astrocytes (the predominant cell type in the human brain), may exert an active role in the pathogenesis of traumatic brain injury (TBI). Activated astrocytes may contribute to increased oxidative stress and neuroinflammation following neurotrauma. Interestingly, the drug Edaravone (also mentioned above—see 4 paragraphs back) has been used successfully, in past research, due to its neuroprotective and antioxidative effects on the brain after TBI. Wang et al. [44] extended this research and found that, after intravenous administration (in rats), Edaravone treatment significantly decreased hippocampal neuron loss, reduced oxidative stress, and decreased neuronal programmed cell death as compared with control treatment. The protective effects of Edaravone treatment were also related to the pathology of TBI on non-neuronal cells, as Edaravone decreased both astrocyte and microglia activation following TBI. These authors conclude that the likely mechanism of Edaravone’s neuroprotective effect in the rat model of TBI is via inhibiting oxidative stress, leading to a decreased inflammatory response and decreased glial activation, and thereby reducing neuronal death and improving neurological function [44]. Similarly, Itoh et al. [45] have reported that intravenous Edaravone administration (in rats), following TBI, inhibited free radical-induced neuronal degeneration and apoptotic cell death around the damaged area. Hence, Edaravone treatment improved cerebral dysfunction following TBI, suggesting its potential as an effective clinical therapy [45]. 

In view of the above description of TBI, the effects of the drug Edaravone, and the pathophysiological overlap of TBI with many characteristics of Alzheimer’s disease brain (cf. above), it is logical and consistent that Jiao et al. [18] have recently reported that Edaravone can also ameliorate Alzheimer’s disease-type pathologies and cognitive deficits of a mouse model of Alzheimer’s disease. Specifically, besides reducing amyloid-β deposition and tau hyperphosphorylation, Edaravone was found to alleviate *oxidative stress* and, hence, attenuates the *downstream pathologies* including glial activation, neuroinflammation, neuronal loss, and synaptic dysfunction, and rescues the memory deficits of the mice [18]. (Note that Edaravone is a small-molecule drug, which is known to function as a free-radical scavenger; it currently is being used clinically in Japan to treat (acute ischemic) stroke patients [18,44].) Jiao et al. [18] further state that their above findings suggest that Edaravone is a promising drug candidate for Alzheimer’s disease by targeting multiple key pathways of the disease pathogenesis. This recommendation by Jiao et al. of Edaravone (for treating Alzheimer’s disease) fits well with the initial drug candidates suggested, based on low-molecular-weight and sufficient lipophilicity, for incorporation into the LCM/ND lipid nanoemulsion proposed here (cf. above) to treat Alzheimer’s disease. Since their recommendation is based in part on knowledge of failed clinical trials indicating that a single target or pathway does not work on this complex disease [18], these investigators are understandably encouraged by a drug like Edaravone which targets multiple pathways of Alzheimer’s disease pathogenesis. 

Another drug candidate suggested above for incorporation into the LCM/ND nanoemulsion is docosahexaenoic acid, or DHA. It has recently been reported extensively, in numerous publications by various groups of investigators worldwide (e.g., [26,27,28,29,30,31,32,33,34]), that DHA has been used successfully to treat Alzheimer’s symptoms in humans as well as animal models (and brain injury in animal models). (See also below.) 

## 7. Targeted Delivery (of Drugs Including Antibody Therapeutics) Coordinated with Focused Sonoporation

More generally, this overall nanotherapeutic approach to treating Alzheimer’s disease, via lipid (LCM/ND)-nanoemulsion particles, is in harmony with the conclusions of a recent review on drug targeting to the brain [46]. Of particular interest, Mahringer et al. [46] point out that one noninvasive approach to overcome the blood–brain barrier has been to increase lipophilicity (even further) of CNS drugs by the use of colloidal drug-delivery carriers, e.g., surfactant/lipid-coated (polymeric) nanoparticles. These authors explain that, after intravenous injection, these surfactant-treated nanoparticles apparently bind to apolipoproteins (e.g., apoA-I in blood plasma) and are subsequently recognized by the corresponding lipoprotein receptors, namely, SR-BI type scavenger receptors at the BBB ([46]; cf. Section 25.2 in ref. [9]). In addition, Mahringer et al. [46] further point out in their review that focused-ultrasound/microbubble (FUS/M) delivery of a model drug has been achieved in the past with minimal histological damage, while demonstrating markedly increased brain dosage (compared to background BBB “leak”), in transgenic Alzheimer’s-disease mouse models [47]. Moreover, in another related study, the FUS/M strategy opened the BBB sufficiently to allow passage of compounds of at least 70 kDa (but not greater than 2000 kDa) into the brain parenchyma. This noninvasive and localized BBB-opening (i.e., sonoporation) technique could, therefore, provide an applicable mode to deliver nanoparticles of a range over several orders of magnitude of daltons [46,48]. As specifically concerns antibody therapeutics, a very recent review [49] cites a published example where dopamine receptor-targeted antibodies could cross the BBB following FUS/M delivery. Also, intravenous (iv) injection of anti-amyloid-β antibodies were observed to cross the BBB following FUS/M delivery and, furthermore, significantly reduced amyloid-β plaques (4 days) post-treatment in a transgenic mouse model of Alzheimer’s disease [50,51]. 

Even without employing sonoporation, Mahringer et al. [46] emphasize that brain uptake of large peptides like lipoproteins is mediated by endocytosis and/or transcytosis through peptide-specific receptors (e.g., scavenger receptors) which are now studied as target moieties for antibody-conjugated nanocarriers. Currently developed CNS drugs include large, hydrophilic molecules like antibodies; while approximately 100% of large molecules ordinarily do not cross the BBB, such large molecules (e.g., antibodies) do in fact pass the membrane barrier when delivered via receptor-mediated endocytosis. As Mahringer et al. [46] point out in their detailed review, the BBB is equipped with several endocytotic receptors at the luminal surface (i.e., capillary endothelial membrane), including the type BI scavenger receptor (SR-BI). These reviewers state that coated nanoparticles represent one of the most innovative noninvasive approaches for drug delivery to the CNS; an important aspect for the commercial development of such nanoparticle systems is the fact that some of the materials employed have already been registered for parenteral use. The authors also cite work published in the past decade (consistent with separate Cav-Con, Inc.-collaborative studies published in the 1990s, see www.netplex.net/~cavcon), using fluorescent-labeled coated nanoparticles and confocal laser scanning microscopy, which provide direct evidence that the (polymer)-coated nanoparticles crossed the BBB and distributed in the brain tissue after iv administration to rats [46]. 

Very recently, the same coated-microbubble approach has been successfully utilized by Mulik et al. [52] for the targeted delivery of a particular therapeutic agent, namely docosahexaenoic acid (DHA), into the brain. Specifically, lipoprotein nanoparticles reconstituted with DHA were employed due to the likelihood of their significant therapeutic value in the brain, since DHA is known to be neuroprotective [52]. Temporary, noninvasive BBB opening was achieved by Mulik et al. using pulsed ultrasound exposures in a localized brain region in normal rats, after which the (fluorescent-labeled or) DHA-containing lipoprotein nanoparticles were administered intravenously. Fluorescent imaging of the rat brain tissue demonstrated that DHA was incorporated into the brain cells (and metabolized) in the ultrasound-exposed hemisphere. In addition, histological evaluation did not indicate any evidence of increased tissue damage in the ultrasound-exposed brain regions compared to normal brain. The authors concluded that their study demonstrates that localized delivery of DHA to the brain is possible using systemically-administered lipoprotein nanoparticles combined with pulsed focused ultrasound exposures in the brain [52]. Other related nanoemulsion formulations for delivery of DHA have also been described recently [53].

Note that (microbubble-assisted) sonoporation not only facilitates localized drug delivery (cf. above) but *also the removal* of amyloid-β plaques from brain tissue in a mouse model [1]. The mechanism of this plaque-burden reduction by sonoporation is believed to involve “loosening the tight junctions of the cells forming the BBB” (see below); at the same time, it is worth noting that this same mechanism might also function to *counteract* characteristic *decreased* “brain clearance” of neurotoxic amyloid-β “monomer” which has been described [54] as a central event in the pathogenesis of Alzheimer’s disease (cf. [55]). Namely, the recent biomolecular study by Keaney et al. reports that controlled modulation of tight junction components at the BBB can *enhance* the clearance (into the plasma) of soluble human amyloid-β monomers from the brain in a murine model of Alzheimer’s disease [54]. 

Lastly, very recently published work (in human-endothelial-cell monolayer cultures as well as in three-dimensional tissue-engineered human vessels) has demonstrated in detail [56] that HDL, acting via scavenger receptors (class B type I, i.e., SR-BI), blocks amyloid-β uptake into endothelial cells—in experimental monolayers as well as, the authors argue, in the human cerebrovascular endothelium [56]. (These authors also point out that SR-BI is the principal HDL receptor on (human brain microvascular) endothelial cells and activates several HDL-signaling pathways (in addition to mediating selective cholesterol uptake) upon HDL docking. The authors observed that inhibiting SR-BI binding with a specific blocking antibody *abolished* the ability of HDL to suppress amyloid-β-induced monocyte adhesion to (human microvascular) endothelial cells. Selectivity to SR-BI was confirmed by demonstrating that blocking scavenger receptor CD36 with a specific antibody had *no* effect on the ability of HDL to suppress amyloid-β-mediated monocyte adhesion to endothelial cells [56]). Also in 2017, Fung et al. separately report that SR-BI [57] mediates the uptake and transcytosis of HDL in brain microvascular endothelial cells (i.e., across the blood–brain barrier). These investigators further argue that manipulation of HDL transcytosis across the blood–brain barrier to increase delivery of apoA-I may, in turn, facilitate increased transport of *“HDL-like synthetic particles”* containing therapeutic drugs across the blood–brain barrier to treat neurodegenerative disorders such as Alzheimer’s disease [57]. Since SR-BI has already been identified as a major receptor for HDL (with their major apolipoprotein *[apo] A-I) as well as for* the earlier-described *LCM/ND nanoemulsion*, this multitasking lipid nanoemulsion can arguably serve as a targeted, apoA-I-based, (SR-BI mediated) therapeutic agent for Alzheimer’s disease [56,58,59,60] (cf. [61,62,63,64,65,66,67]).

## 8. Further Details on Sonoporation across the Endothelial-Cell Monolayer of the BBB

The characteristic LCM subpopulation (i.e., a stable (>6 months) LCM suspension contained within the LCM/ND-nanoemulsion total particle population) would now be available to substantially reduce the acoustic power levels needed for accomplishing endothelial sonoporation (cf. [68]), if additionally desired for further targeted neurotherapy of the Alzheimer’s patient. Over the past decade, neuroscientists have been exploring the use of ultrasound and preformed microbubbles to temporarily open the BBB [69,70,71,72,73,74], allowing drugs or the immune system to target brain tumors or Alzheimer’s brain plaque. It is believed that (non-thermal focused) ultrasound pulses cause the (intravenously injected) preformed microbubbles to expand and contract (with acoustic pressure rarefaction and compression, respectively) against the BBB structure, thereby loosening the tight junctions [1,2] between endothelial cells which form the structural core of the BBB. Recently, this research approach was employed by Leinenga and Gotz [1] who utilized focused ultrasound coupled with intravenous injection of lipid-encased microbubbles. Their procedure design was sufficient to both remove amyloid-β plaques in a mouse model of Alzheimer’s disease, in which amyloid-β is deposited in the brain, and to restore memory function in mice with Alzheimer’s-like symptoms [1]. More specifically, these investigators report that repeated scanning ultrasound (SUS) treatments of the mouse brain activated the invading microglia—which, in turn, caused extensive internalization of amyloid-β plaques into the lysosomes of activated microglia following the SUS treatments. Besides reducing plaque burden, the SUS-treated mice also displayed improved performance on three memory tasks (the Y-maze, the novel object recognition test, and the active place avoidance task). Leinenga and Gotz conclude that their findings suggest that repeated SUS is useful for removing amyloid-β plaques in the mouse brain without causing observable damage, and should be explored further as a noninvasive method with potential as a (non-pharmacological) therapeutic approach for Alzheimer’s disease [1]. 

The actual cellular and biophysical mechanism(s) of the reversible BBB “opening” process from sonoporation, when employing focused transcranial ultrasound coupled with injected preformed microbubbles, has been described further in other published studies over the last several years. For example, the preformed microbubbles concentrate the ultrasound effects to the microvasculature, greatly reducing the ultrasound exposure levels needed to produce bioeffects; thus, with injected microbubbles one can apply focused ultrasound transcranially without significant skull heating [75,76]. Note that the term sonoporation has been used, in the literature, for describing both cell-membrane as well as blood-vessel permeabilization. Also, the repeated expansion and compression of microbubbles upon exposure to low acoustic pressures is referred to as stable, or non-inertial, cavitation [77]. Because the circulating microbubbles appear to concentrate the ultrasound effects to the blood vessel walls, the temporary opening of the BBB occurs through the widening of tight junctions between endothelial cells (and possibly also activation of transcellular mechanisms) with little effect on the surrounding brain parenchyma [75,76,77]. This observed increase of BBB permeability is transient (i.e., the induced opening of the BBB is reversible) and apparently safe [75,76], which is further supported by related cell-culture studies demonstrating viability of endothelial cells after ultrasound/microbubble-mediated sonoporation [78,79]. In particular, using human (umbilical vein) endothelial cell (HUVEC) monolayer cultures [78], it was found that ultrasound/microbubble-mediated sonoporation results in observable acute cellular-pore generation (resealing time < 2 min) and, more importantly, generates intercellular gaps between adjacent confluent HUVECs that persist over longer timescales (~30–60 min). Confocal-microscopy/cell-viability assays repeatedly confirmed that the cultured endothelial cells remained viable at 40 min post-ultrasound transmission, suggesting a visible mechanism for these authors’ findings of prolonged, enhanced vascular permeability [78]—again documenting that the ultrasound/microbubble-mediated opening of the BBB can be considered temporary, reversible, and apparently safe [75,76,77,78,79].

Moreover, other investigators have recently pointed out [80,81] that microbubble-mediated sonoporation is also believed to actually enhance local drug uptake across the cell membrane itself (e.g., of endothelial cells); namely, besides transient cellular-pore generation, the ultrasound-induced oscillation of local (preformed) microbubbles causes cell-membrane deformation, which is hypothesized to induce or facilitate endocytosis [80,81]. More specifically, real-time confocal microscopy recording (during *lower*-acoustic-pressures ultrasound application) revealed that the membrane deformation by oscillating microbubbles may be the trigger for endocytosis via mechanostimulation of the cytoskeleton [80]. Hence, CNS-endothelial sonoporation offers a range of neurotherapeutic options that can include either: (1) inducing/facilitating endocytosis (and, in turn, transcytosis); (2) transient cellular-pore generation; and/or (3) widening of tight junctions between endothelial cells of the cerebral microvasculature. These varied neurotherapeutic options are important and useful for both the researcher and the clinician, because the BBB disruption associated with various neurological disorders (e.g., Alzheimer’s disease, vascular dementia) has not been characterized in full detail cellularly. Namely, while the endothelial cells that form the structural core of the BBB normally have specialized tight junctions and extremely low rates of transcytosis (to limit flux of substances between the blood and CNS) [82,83,84], the relative contributions of these altered properties to the disrupted BBB permeability in different neurodegenerative disorders (e.g., Alzheimer’s disease) is unclear or unknown [84]. In the foreseeable future, taking full advantage of this proposed minimally invasive and targeted use of preformed (LCM/ND nanoemulsion-based) microbubbles with sonoporation, while optimizing drug-delivery efficiency (through judicious choice of acoustic parameters) and minimizing side effects, may assist in advancing sonoporation to the clinic (cf. [80,81,85,86]).

In this neurotherapeutic approach to the clinic, both the researcher and the clinician are still faced with the challenge of translation from rodent to large animal or man—yet significant progress on minimizing potential side effects, in large-animal transcranial-ultrasound work, has already been reported in the literature. For example, an earlier study by Xie et al. [76] in pigs has demonstrated that intravenous lipid-encapsulated microbubbles, combined with transtemporal-applied 1 MHz ultrasound, can transiently and reversibly increase BBB permeability in a large-animal model. These authors explain that the peak negative pressure *decrease* across the pig temporal bone was nearly ten-fold with the 1 MHz probe used for their study. Despite this attenuation, they were still able to alter BBB permeability [76]. Xie et al. further point out that the duration of altered BBB permeability was also less than what has been observed with higher peak negative pressures applied (in published reports by other investigators they cite) through bone windows in smaller animals. While other ultrasound parameters (pulse length, duty cycle) may play a role in altering permeability, this degree of attenuation in peak negative pressure may largely determine how effective transcranial ultrasound is as a noninvasive method in transiently altering blood–brain barrier permeability for drug delivery. Xie et al. conclude that their study achieved an alteration in BBB permeability with lower peak negative pressures and lower doses of ultrasound contrast (i.e., intravenous, film-stabilized microbubbles) in a large-animal model and, thus, transient alterations in BBB permeability sufficient for drug delivery and without unwanted bioeffects (hemorrhage, necrosis, apoptosis) [76] appear increasingly feasible.

## 9. Conclusions

By incorporating drug candidates (such as Edaravone, DHA, or antibody therapeutics) into the LCM/ND lipid nanoemulsion type, known to be a successful drug carrier [9], one is likely to obtain a multitasking combination therapeutic for translational medicine. This therapeutic agent would target cell-surface SR-BI making it possible for various (above-described) cell types, all potentially implicated in Alzheimer’s disease (cf. [87,88]), to be simultaneously sought out and better reached for localized drug treatment of brain tissue in vivo. Further, it has been reconfirmed in the current literature that receptor-mediated endocytosis/transcytosis via lipoprotein receptors, particularly scavenger receptors including SR-BI, remains a major route for drug delivery across the blood–brain barrier; namely, recently published work has demonstrated that nanocomplexes can be readily transported into brain capillary *endothelial* cells (bovine and porcine) via SR-BI receptor-mediated endocytosis ([89]; see also [90,91,92]). Accordingly, *endothelial* modulation and repair become feasible by pharmacological targeting [93,94,95,96,97,98,99,100,101] via SR-BI receptors (cf. [102]). Moreover, the effects of the various cell types targeted (via SR-BI) may be additive, multiplicative, or otherwise synergistic. This therapeutic approach receives added impetus from continual findings of cerebrovascular pathology [103,104,105,106,107,108,109] and brain arterial aging [110,111,112,113] accompanying, and an apparent *endothelium*-dysfunction involvement [93,94,95,96,97,98,99,100,101,109,114,115,116,117,118,119,120] in, both Alzheimer’s disease and its major risk factors [113,114,115,116,117,118,119,120,121,122,123,124,125,126,127,128,129,130]. Hence, the proposed multitasking combination therapeutic may also display greater effectiveness at different stages of Alzheimer’s disease (cf. [87,88]); as a result, this multitasking (drug-delivery) therapeutic could represent a promising way to treat, delay, or even prevent the disease in the future. Lastly, a completely separate and additional advantage of such LCM/ND lipid nanoemulsion(s), as a component of this combination therapeutic, stems from the characteristic lipid-coated microbubble subpopulation [9] existing in this nanoemulsion type. Specifically, such preformed (lipid-stabilized) microbubbles are well known to substantially reduce the acoustic power levels needed for accomplishing temporary noninvasive (transcranial) ultrasound treatment [4,5,6,7,8,131,132,133,134], or sonoporation [135,136,137,138,139,140,141,142], if additionally desired for the Alzheimer’s patient.

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
