# Peer review of "Alzheimer’s Disease, Brain Injury, and C.N.S. Nanotherapy in Humans: Sonoporation Augmenting Drug Targeting"

_medsci, 2017, doi:10.3390/medsci5040029_

Round 1

Reviewer 1 Report

SRB1 is highly expressed in liver sinusoidal endothelial cells. please discuss relevance for biodistribution of LCM/ND.

is there any direct evidence for involvement of SRB1 receptor in targeting of LCM/ND? for example, by knockdown of SRB1 and/or blocking of LCM/NDinteraction with SRB1 by HDL/LDL? this literature should be discussed.

the fact that only lipophilic drugs are delivered using LCM/ND may reflect passive leakage of the drug from the carrier system instead of active transport through endocytic events. please discuss.

if SRB1 mediates uptake of nanocarriers by endothelial cells, what will happen with the LCM/ND at the BBB? what is the size of the LCM/ND system? will it be internalized by ECs before sonoporation can create openings at the BBB that will allow for passage into brain tissue?

line 74, is reference 9 the correct reference?

 the author often refers to specific sections and figures/tables of reference 9. As this paper is not freely accessible, it may be preferred to ask for permission to reproduce some of these figures/tables in the current paper.

Author Response

Reply to Review #1:

Thank you for your detailed review.

1) SRB1 is highly expressed in liver sinusoidal endothelial cells. please discuss relevance for biodistribution of LCM/ND.

2) Is there any direct evidence for involvement of SRB1 receptor in targeting of LCM/ND? for example, by knockdown of SRB1 and/or blocking of LCM/NDinteraction with SRB1 by HDL/LDL? this literature should be discussed.

ANSWER re: paragraphs 1&2:

High expression of SR-BI in the liver, and its relevance to the biodistribution of LCM/ND, is mainly treated in the Section entitled “Serum Amyloid A (SAA), SR-BI, and Alzheimer's Disease” (with refs. therein).  Briefly, the liver is the major site of (HDL-associated) acute-phase-protein SAA expression, but SAA is also expressed in cells at inflammation sites.  SR-BI functions as an endocytic SAA receptor and, accordingly, is also highly expressed on monocytes/macrophages -- which are known to be primary sites of SAA uptake.  A recent study reported that such blood-borne human monocytes were found to play a central role in ameliorating cognitive deficits and reducing β-amyloid neuropathology in an Alzheimer's mouse model.  As concerns “knockdown of SR-BI”, an earlier study (by different investigators) reported that SR-BI knockout mice displayed impairments in recognition memory and spatial memory.  Again regarding SAA and inflammation interrelationships, inflammation is a known risk factor for Alzheimer's disease and the SAA concentration is much higher, in the cerebrospinal fluid, in subjects with Alzheimer's disease than in controls.

3) he fact that only lipophilic drugs are delivered using LCM/ND may reflect passive leakage of the drug from the carrier system instead of active transport through endocytic events. please discuss.

ANSWER re: paragraph 3:

Active transport of LCM/ND nanoemulsion via receptor-mediated endocytic events has been reported earlier, elsewhere in the peer-reviewed literature.  Briefly, the earlier experiments involved three types of measurements: kinetics of uptake, temperature/energy dependence, and confocal microscopy using dual-channel recording (J. Neuro-Oncol. 26 (1995) 25-34).  For example, most 9L and C6 brain-tumor cells examined appear to have LCM in their cytoplasm 2 min after LCM i.v. injection in rats.  This time course of internalization and temperature dependency of the process are consistent with endocytosis, as is inhibition of LCM uptake by energy blockers, since endocytosis is a temperature-dependent active-uptake process.  In addition, approximately 70% of the total measured diO-labeled LCM (including agglomerations of the far more numerous and much smaller, diO-labeled lipid nanoparticles [previously undetected] ) internalized by C6 brain-tumor cells were found to be associated with acidic (subcellular) compartments -- which comprise endosomes and/or lysosomes.  These cell-culture data, similar to the above (in vivo) kinetic data, further confirmed that C6 brain-tumor cells take up LCM via an endocytic pathway (Barbarese et al., J. Neuro-Oncol. 26 (1995) 25-34;  D'Arrigo, Stable Nanoemulsions (2011) 436 pp., Elsevier).

4) If SRB1 mediates uptake of nanocarriers by endothelial cells, what will happen with the LCM/ND at the BBB? what is the size of the LCM/ND system? will it be internalized by ECs before sonoporation can create openings at the BBB that will allow for passage into brain tissue?

ANSWER re: paragraph 4:

As described in the first two paragraphs of the Section entitled “Targeted Delivery (of drugs … with Focused Sonoporation” (with refs. therein), the detailed review by Mahringer et al. [46] points out that (after intravenous injection) surfactant/lipid-coated nanoparticles apparently bind to apolipoproteins (for example, apoA-I in blood plasma) and are susequently recognized by the corresponding lipoprotein receptors, namely, SR-BI type scavenger receptors at the BBB ( [46]; cf. Sect. 25.2 in ref. [9] ).  These authors also cite work published in the past decade (and consistent with separate Cav-Con,Inc.-collaborative studies published in the 1990s [see www.netplex.net/~cavcon] ), using fluorescent-labeled coated nanoparticles and confocal laser scanning microscopy, which provide direct evidence that the [polymer-]coated nanoparticles crossed the BBB and distributed in the brain tissue after i.v. administration to rats [46].  [As specifically concerns the LCM/ND system, LCM are mostly submicron while the ND are far more numerous and much smaller (under 0.2 μm [i.e., under 200 nm] in diameter).]  Related, further discussion of SR-BI, HDL, and β-amyloid interactions at the endothelial-cell monolayer (i.e., BBB core) is presented in the last paragraph of this Section (in the REVISED manuscript), -- while many added details concerning the sonoporation process (across the endothelial-cell monolayer/BBB) are presented in the following text Section (of the REVISED manuscript).

5) line 74, is reference 9 the correct reference?

ANSWER re: paragraph 5:

In (the original) line 74, the reference citation was correct but (in the REVISED manuscript) now more appropriately reads as:  “(for a review, see Sect. 24.2 in ref. [9] )”.

 6) The author often refers to specific sections and figures/tables of reference 9. As this paper is not freely accessible, it may be preferred to ask for permission to reproduce some of these figures/tables in the current paper.

ANSWER re: paragraph 6:

Regarding figures/tables of ref. [9], I refer only twice now to a Table 24.1 (in the fourth Section [“Past Targeted …” ], paragraph 1).  In each of these two cases (in the REVISED manuscript), I have added a citation to the corresponding peer-reviewed journal source (with exact page no.) of the “parent table” from which Table 24.1 was adapted.

Reviewer 2 Report

Thank you for writing this review. It is clearly a review to promote sonoporation which I appreciate but it will require significant caveats and additions prior to publication as same.

Yes, the ideas presented on liposome and other ideas of treating multivariate aspects of any CNS disease such as ALZ is excellent and understandable. While many ALZ experts may disagree that the many cell types effected by ALZ may be impossible to treat the ideas and concepts presented in this review are worth discussion in a open and transparent manner.

Transparency on delivery is the concern with this review. Yes, RMT and RMD both work when specific receptors are focused but not all endothelial cell receptors can be used and this must be presented as caveats in this review . Too much outstanding work on RMT has been presented to be ignored as to the need for RMT specificity.

Transparent open discussion (in depth) on the concerns and lack of human or primate or large animal sonoporation work must be presented in a separate section so as to educate the reader in the reasons for the lack of translation from rodent to man and the dangers associated with sonoporation for man today. This is very important as this delivery technique may some day work if the writer of this review can educate all readers as to the challenges in terms of energy applied, skull thickness, skull biometrics of man versus rodent. How can this challenge be addressed needs to be at the focus of their review or the excellent ideas of lipid based delivery can not happen.

Last of all is the short but appreciated discussion on BBB tight junctions in the manuscript today. This is a area of great interest to many drug delivery experts because  disruption occurs in several disease states but death does not follow....i.e. the tight junctions remain. This is proof of the rigorous nature of the BBB TJ. It is real proof of concept .  The TJ of the BBB is such a dynamic set of proteins.  Each TJ protein with signaling response elements communicating with other Neurovascular Unit cell types.  Yes,  short term disruption ( sonoporation)  in a very serious disease state such as GBM or even ALZ may  be dramatic but also it may be all we have to work with in a given situation. If the author can address all of this the review would be much better and well read by many in and out of the field.

Author Response

Reply to Review #2:

Thank you for your detailed review.

ANSWER re: paragraphs 1-3:

As described in the first two paragraphs of the Section entitled “Targeted Delivery  (of drugs …” (with refs. therein), the detailed review by Mahringer et al. [46] emphasizes that (even without employing sonoporation) brain uptake across the BBB of large molecules, like lipoproeins, occurs via receptor-mediated endocytosis and/or transcytosis (RMT).  Mahringer et al. also point out that the BBB is equipped with several endocytic receptors at the luminal surface (i.e., the capillary endothelial membrane), including the type BI scavenger receptor (SR-BI).  These authors explain that, after i.v. injection, surfactant/lipid-coated nanoparticles apparently bind to apolipoproteins (for example, apoA-I in blood plasma) and are subsequently recognized by the corresponding lipoprotein receptors, namely (when in the presence of apoA-I ), SR-BI type scavenger receptors at the BBB.  These authors also cite work published in the past decade (and consistent with separate Cav-Con, Inc.-collaborative studies published in the 1990s [see www.netplex.net/~cavcon] ), using fluorescent-labeled coated nanoparticle and confocal laser scanning microscopy, which provide direct evidence that the [polymer-]coated nanoparticles crossed the BBB and distributed in the brain tissue after i.v. administration to rats [46].  Related, further discussion of SR-BI, HDL, and β-amyloid interactions at the endothelial-cell monolayer/BBB (including in Alzheimer's disease) is presented in the last paragraph of this Section (in the REVISED manuscript).

ANSWER re: paragraphs 4&5:

Many added details on the sonoporation process (across the endothelial-cell monolayer/BBB) are presented in the next Section (of the REVISED manuscript) entitled “Further Details on Sonoporation across the ...” (with numerous [many quite recent] refs. therein) to assist readers in advancing sonoporation to the clinic.

Round 2

Reviewer 1 Report

no further comments.

Reviewer 2 Report

Author has answered the questions. This is an interesting review.

Med. Sci. EISSN 2076-3271 Published by MDPI AG, Basel, Switzerland RSS E-Mail Table of Contents Alert
Back to Top